# Combination of Talazoparib and Palbociclib as a Potent Treatment Strategy in Bladder Cancer

**DOI:** 10.3390/jpm11050340

**Published:** 2021-04-24

**Authors:** Florian G. Klein, Charlène Granier, Yuling Zhao, Qi Pan, Zhichao Tong, Jürgen E. Gschwend, Per Sonne Holm, Roman Nawroth

**Affiliations:** 1Department of Urology, Klinikum Rechts der Isar, Technical University of Munich, D-81675 Munich, Germany; florian.gerhard.klein@tum.de (F.G.K.); charlene.granier@tum.de (C.G.); yuling.zhao@tum.de (Y.Z.); qi.pan@shgh.cn (Q.P.); zhichao.tong@hrbmu.edu.cn (Z.T.); juergen.gschwend@tum.de (J.E.G.); per-sonne.holm@i-med.ac.at (P.S.H.); 2Department of Oral and Maxillofacial Surgery, Medical University Innsbruck, A-6020 Innsbruck, Austria

**Keywords:** bladder cancer, combination therapy, Talazoparib, Olaparib, Palbociclib, Retinoblastoma protein, apoptosis, chorioallantoic membrane model

## Abstract

The use of cyclin-dependent kinase 4/6 (CDK4/6) inhibitors represents a potent strategy for cancer therapy. Due to the complex molecular network that regulates cell cycle progression, cancer cells often acquire resistance mechanisms against these inhibitors. Previously, our group identified molecular factors conferring resistance to CDK4/6 inhibition in bladder cancer (BLCA) that also included components within the DNA repair pathway. In this study, we validated whether a combinatory treatment approach of the CDK4/6 inhibitor Palbociclib with Poly-(ADP-Ribose) Polymerase (PARP) inhibitors improves therapy response in BLCA. First, a comparison of PARP inhibitors Talazoparib and Olaparib showed superior efficacy of Talazoparib in vitro and displayed high antitumor activity in xenografts in the chicken chorioallantoic membrane (CAM) model. Moreover, the combination of Talazoparib and the CDK4/6 inhibitor Palbociclib synergistically reduced tumor growth in Retinoblastoma protein (RB)-positive BLCA in vitro and in a CAM model, an effect that relies on Palbociclib-induced cell cycle arrest in G0/G1-phase complemented by a G2 arrest induced by Talazoparib. Interestingly, Talazoparib-induced apoptosis was reduced by Palbociclib. The combination of Palbociclib and Talazoparib effectively enhances BLCA therapy, and RB is a molecular biomarker of response to this treatment regimen.

## 1. Introduction

Small-molecule inhibitors against cyclin-dependent kinase 4/6 (CDK4/6) are potent compounds for cancer treatment [1]. CDK4/6 are kinases that regulate the cell cycle transition from G1- into S-phase. A recent comprehensive analysis of different molecular subtypes in a cohort of 412 muscle-invasive bladder cancers (BLCA) revealed that approximately 90% of these tumors displayed genetic alterations in regulatory elements of the cell cycle pathway [2]. In normal cells, the Retinoblastoma protein (RB), forming the pocket protein family with p107 and p130, interacts with and regulates the activity of the E2F transcription factor family [3]. Upon phosphorylation by CDK4/6, E2F is released from the RB/E2F inhibitory complex, thus activating the transcription of E2F-regulated proteins that initiate the cellular transition from G0/G1- to S-phase [4]. The antiproliferative potential of the CDK4/6 inhibitor Palbociclib in RB-positive BLCA cells has been shown [5,6]. However, due to the complex network of cellular agonists and antagonists that regulate the activity of these kinases, CDK4/6 inhibitors as monotherapy performed poorly in most clinical trials [3,7,8]. A first phase-2 clinical trial in BLCA investigating the CDK4/6 inhibitor Palbociclib was recently terminated, as the primary endpoint from the first 12 subjects could not be met [9]. An early phase-1 trial (NCT03837821) is currently evaluating the clinical potential of the CDK4/6 inhibitor Abemaciclib as monotherapy in BLCA.

The lacking effectiveness of CDK4/6 inhibitors as monotherapies in clinical trials led to the identification and development of novel combination therapies [7]. In a gain-of-function genome-scale CRISPR-dCas9 screen, our group identified molecules and signaling pathways that confer resistance to Palbociclib treatment [10]. Among them, genes involved in key processes of DNA repair mechanisms were identified. This suggested that the combination of CDK4/6 inhibitors and inhibitors against DNA repair mechanisms might improve response to Palbociclib monotherapy. Major molecular components of these pathways are members of the Poly (ADP-Ribose) Polymerase (PARP) protein family. Small-molecule inhibitors against PARP have been developed and approved by the Food and Drug Administration (FDA) for the treatment of ovarian or prostate cancer with the inclusion criteria of germline BRCA1/2 mutation [11,12]. In consequence, we further examined whether the combination of PARP inhibitors and CDK4/6 inhibitors might function as a supportive mechanism for the treatment of BLCA.

PARP acts as an essential molecule in the cellular DNA damage response [13]. The pharmacological inhibition of PARP-1 abolishes DNA single-strand repair and traps the enzyme at sustained single-strand breaks, converting them into DNA double-strand breaks (DSB) upon disruption with processive replication forks [14]. This accumulation of DSBs triggers the γ-phosphorylation of damage response Histon-2A (γH2A.X), leading to the subsequent activation of Checkpoint Kinases 1/2 (Chk-1/2) and finally to p53-mediated apoptosis [15]. DSBs can be repaired via template-guided Homologous Recombination (HR) or error-prone Nonhomologous End-Joining (NHEJ) pathways. HR-factors such as BRCA1/2 or RAD51 are common tumor suppressor genes that are frequently mutated in cancers, thereby providing the opportunity for synthetic lethal treatment regimens with PARP inhibitors [16]. To date, all FDA-approved PARP inhibitors share the same mode of action as NAD^+^ competitors at its PARP-1 substrate binding site. Among them, Olaparib was the first to receive FDA approval. Talazoparib represents the most recent generation of PARP-1 inhibitors with the highest affinity to its target [13,17]. However, resistance phenotypes have been observed during prolonged treatment with PARP inhibitors due to the restoration of HR, increased PARP-1 activity, or induced clearance of the drug (reviewed in detail in [18]). Several clinical trials are currently investigating the PARP inhibitors Olaparib (NCT03459846, NCT02546661, NCT04579133, NCT03375307, NCT03448718) and Niraparib (NCT03945084, NCT03425201) in bladder or urothelial cancer as monotherapies or in combination with immune checkpoint blockade.

Because the clinical benefit of CDK4/6 or PARP-1 targeted inhibition as respective monotherapies might be, as also shown in other tumor entities [19,20], limited for BLCA, suitable combination therapies for these compounds are necessary for sufficient therapeutic efficiency. Therefore, we characterized the value of combining these two classes of targeted inhibitors for the treatment of BLCA and evaluated potential biomarkers determining its scope of application.

## 2. Results

### 2.1. Expression of Genes Involved in the DNA Repair Pathway Confer Resistance to CDK4/6 Inhibitors

In a genome-scale CRISPR-dCas9 activation screen, our group identified markers of resistance to the CDK4/6 inhibitor Palbociclib in BLCA [10]. This screen detected 995 genes that were further analyzed for their involvement in functional signaling pathways using Reactome analysis (Figure 1A). Three out of four genes belonging to the “resolution of apurinic/apyrimidinic site via single-nucleotide replacement pathway” were significantly enriched, indicating the involvement of DNA repair pathways in therapy response (Figure 1B). The initial sensor of DNA lesions in the single-nucleotide replacement pathway is the PARP-1 enzyme (Figure 1C). In this project, we tested the combination of PARP inhibitors and Palbociclib as a novel therapy approach in BLCA that might overcome acquired resistance mechanisms against CDK4/6 inhibitors.

### 2.2. Talazoparib Exhibits Superior Efficacy Than Olaparib against BLCA

Because most studies with PARP inhibitors in BLCA have been performed using Olaparib, we first aimed to compare the potency of the latest generation PARP inhibitor Talazoparib to Olaparib in dose-response assays (Figure 2). Analyzing the reduction of cell viability in a panel of BLCA cell lines, we observed that IC_50_ values of Talazoparib were 20-300-fold lower than those of Olaparib (Table 1). Although the amplitude of this enhancement shows some variations between cell lines, Talazoparib appears to be more efficient, as described in other cancer entities [17]. Therefore, all further experiments in this study we conducted using Talazoparib.

### 2.3. Talazoparib Induces DNA Damage and Apoptosis in BLCA

The suggested mechanism of cytotoxicity via PARP inhibition is the accumulation of DNA damage and the subsequent induction of apoptosis [21,22]. When applying whole-cell lysates of Talazoparib-treated cells to Western blot analysis, we observed a significant increase in the expression of Chk-1 and γH2A.X, indicating the upregulation of cellular damage response mechanisms (Figure 3A). The activation of apoptosis was further reinforced through an increase in cleaved fragments of Caspase-3 and PARP (Figure 3B). We functionally confirmed this phenomenon by detecting Caspase-3/7 activation in a *D*-Luciferin-based metabolic assay (Figure 3C).

### 2.4. Talazoparib Effectively Suppresses Tumor Growth in a BLCA Model

The IC_50_ concentration of Talazoparib in RT-112 cells was shown to be approximately 200 nM (Table 1). To further verify the cytotoxic potential of Talazoparib against BLCA in a three-dimensional xenograft model, we generated tumor xenografts using RT-112 cells (as this cell line forms large and highly vascularized tumor xenografts on the CAM) stably expressing luciferase and implemented them on the chicken chorioallantoic membrane (CAM) [23]. We used this model because it represents a suitable intermediate stage between isolated cultured cells and animals and is in line with the 3R-guiding principle to replace, reduce, and refine the use of animals in scientific research. The treatment of xenografts with 200 nM Talazoparib resulted in a highly significant reduction of tumor growth (80%) five days after treatment (Figure 3D). These data indicate that Talazoparib displays high antitumor activity in BLCA as monotherapy.

### 2.5. Combination of Talazoparib and Palbociclib Displays Synergism

This study aimed to characterize the value of combining a PARP inhibitor with a CDK4/6 inhibitor for cancer therapy [10]. Thus, using a cell survival assay, we evaluated the combination of the PARP inhibitor Talazoparib and the CDK4/6 inhibitor Palbociclib in BLCA cell lines. Because the applied BLCA cell lines showed a wide variation in sensitivity to PARP inhibitor treatment, we optimized the dosage of Talazoparib for each cell line individually. We used low concentrations of Talazoparib in the combination and observed synergism in all drug concentrations tested (Figure 4A). Data were analyzed by means of the Chou–Talalay method for drug combination [24]. Based on this analysis, synergism (CI < 1) for the interaction of both drugs was identified with CI values ranging from 0.14 to 0.60 (Figure 4B). In vivo analysis of this combination therapy validated the significant enhancement and showed a significant decrease in the combination therapy compared to both monotherapies (Figure 4C). When analyzing the arithmetic mean of the bioluminescence emitted from living tumor cells in the CAM model, the combination therapy showed a decrease of 41%, whereas the monotherapies led to a decrease of 14% or 12% for Palbociclib or Talazoparib, respectively. It should be noted that we used a low dose of Talazoparib (50 nM) because higher doses already resulted in substantial tumor toxicity in the monotherapy (Figure 3D).

### 2.6. Induction of Apoptosis by Talazoparib Is Reversed in Combination with Palbociclib

We previously showed the induction of apoptosis following Talazoparib monotherapy (Figure 3B). Therefore, we characterized this induction in combination with Palbociclib. Proteolytic cleavage of PARP-1 serves as a biomarker for the detection of programmed cell death. Only Talazoparib monotherapy induced cleavage of PARP-1, whereas Palbociclib reduced the level of cleaved PARP-1 but increased the level of cleaved Caspase-3 in T24 cells. In UM-UC-3 cells, Talazoparib also induced a higher level of cleaved PARP-1 and Caspase-3, whereas Palbociclib did not have any influence on the stability of these enzymes. In combination, Palbociclib suppresses the cell-line-dependent Talazoparib-induced cleavage of PARP-1 and Caspase-3 to different extents (Figure 5A,B). In a functional Caspase-3/7 activation assay, Palbociclib suppressed enzymatic activity by 50–80% compared to untreated samples (Figure 5C). In contrast, Talazoparib induced not only Caspase cleavage efficiently but also led to 150–250% increased activity. Surprisingly, the combination with Palbociclib completely reversed this Talazoparib-induced increase in Caspase-3 cleavage and Caspase-3/7 activation. However, the total protein level of PARP was considerably reduced by monotherapy of Palbociclib and by the combination of both inhibitors (Figure 5A), supporting our initial finding that the activation of the DNA repair mechanism would result in resistance to Palbociclib treatment (Figure 1).

### 2.7. Talazoparib/Palbociclib Combination Therapy Induces Both G0/G1 and G2 Arrest

The induction of cell cycle arrest at the G0/G1 phase upon treatment with CDK4/6 inhibitors is a well-characterized phenomenon [6]. Furthermore, PARP inhibition results in a G2 arrest due to remaining replication-related DNA damage [25]. We examined the respective effect of both therapies by quantifying the expression level of key proteins in cell cycle progression. Talazoparib monotherapy did not affect the RB expression level, whereas the complete downregulation of RB was observed in the Talazoparib/Palbociclib combination (Figure 6A). Talazoparib monotherapy had no effect on Cyclin E2/D1 expression, while treatment with Palbociclib resulted in the downregulation of Cyclin E2 and the upregulation of cyclin D1 in both monotherapy and combination therapy (Figure 6A). The effect of Palbociclib on cell cycle progression was also determined via flow cytometry analysis after EdU incorporation and staining with 7-AAD (Figure 6B). Both, G0/G1 or G2 arrests induced by the respective monotherapy with Palbociclib or Talazoparib could be confirmed. In the combination, although the profound Palbociclib-induced G0/G1 arrest was slightly reduced, this effect correlated simultaneously with a slight increase in cells undergoing S-phase but also enhanced G2 arrest (Figure 6B). These observations might indicate that the synergistic response of the Palbociclib/Talazoparib therapy for BLCA is likely due to a cooperation between the induced G0/G1 and G2 arrest.

### 2.8. Response to Palbociclib/Talazoparib Dual Therapy Depends on the Expression of RB

Response to CDK4/6 inhibitors is determined by the expression of the Retinoblastoma protein (RB) in most cell lines [5]. To examine whether the observed effects in the combination treatment are dependent on RB expression level, we constitutively suppressed RB expression in the muscle-invasive BLCA cell line T24 using RB-directed shRNA (T24shRB). RB suppression was confirmed via Western blot analysis for two different shRNA sequences and three clones each (Figure 7A). Upon monotherapy with Palbociclib, T24shRB cells showed resistance to therapy in cell viability assays and still progressed to S-phase in EdU-incorporation assays (Figure 7B,C). To evaluate the influence of RB depletion for Talazoparib/Palbociclib combinatory therapy, we additionally studied the RB-negative BLCA cell line 5637, which harbors a loss of function mutation in the *RB1* gene [26]. Cell viability studies in 5637 and T24shRB cells revealed CI values ranging from 0.70 to 1.20 for the Talazoparib/Palbociclib combination, which tended to indicate no or additive effects of the inhibitors’ interaction (Figure 7D,E). Palbociclib monotherapy did not affect Caspase-3/7 activity, and the activation of Caspase-3/7 by Talazoparib monotherapy was not reduced in the combination therapy in RB-negative cells (Figure 7F). Collectively, these results demonstrate the crucial role of Retinoblastoma protein expression in mediating therapy response to CDK4/6 inhibitors and indicate that the observed effects within the Talazoparib/Palbociclib combination are essentially dependent on RB expression.

## 3. Discussion

CDK4/6 inhibitors represent a new generation of successful anticancer drugs [7,27]. However, tumor cells can rapidly develop mechanisms of acquired resistance to these inhibitors. Combination with endocrine therapy has already been translated into the clinic and demonstrated the increased therapeutic potential of combining CDK4/6 inhibition and targeted therapies (reviewed in [28]). Thus, inhibitors of the PI3K/Akt/mTOR pathway seem to efficiently counteract molecular adaption to prolonged CDK4/6 inhibitor treatment [29,30]. The combination of CDK4/6 inhibition and chemotherapy has also been shown to be beneficial, although with varying efficiency depending on the tumor entity [5,31,32,33]. Intending to determine genes that confer resistance to the CDK4/6 inhibitor Palbociclib, our group previously performed a genome-wide gain-of-function CRISPR-dCas9 screen and identified the key process of DNA repair [10].

DNA repair pathways are druggable by targeted inhibition of the PARP family of proteins. The capability of trapping PARP-1 defines the drug’s clinical potential, and Talazoparib presents the highest trapping potential resulting from enhanced target affinity with a 3.4-fold higher IC_50_ towards PARP-1 in cell-free assays compared to Olaparib [14,17]. We were able to show that this enhanced target affinity of Talazoparib translates into a significant reduction in tumor cell viability on BLCA. In clinical use, this could lead to a favorable toxicity profile in patients compared to other PARP inhibitors. While Talazoparib has been extensively studied in cancers of ovarian, prostate, or breast that emerge with a high population of germline deficiencies in homologous repair pathways, the effect of Talazoparib in BLCA has only been evaluated on the verge of other research foci. Liu et al. examined the reduced functionality of DNA repair following SA1-depletion in SA2-deficient Ewing sarcoma, and BLCA cell lines and demonstrated a synergistic interaction between SA1-knockdown and a panel of PARP inhibitors [34]. We show here that the combination therapy of the CDK4/6 inhibitor Palbociclib and the PARP inhibitor Talazoparib has synergistic anticancer effects on RB-positive urothelial carcinoma cells in vitro and in xenograft models. Our findings are fortified by recently published studies describing the synergistic effect of CDK4/6 inhibitor Palbociclib and the PARP inhibitors Olaparib or Niraparib against breast cancer cells [35]. Yi et al. reported an improvement in response to monotherapy with Palbociclib when combined with Olaparib in ovarian cancer [36].

Mechanistically, we show here that the observed effects are likely due to a combination of G0/G1 and G2 arrest induced by the two inhibitors. We also demonstrate that the novel PARP inhibitor Talazoparib induces DNA damage and subsequent apoptosis when used as monotherapy after sustained exposition in BLCA cell lines and three-dimensional xenograft models, providing evidence that the reported anticancer activity of Talazoparib should also apply to BLCA. When analyzing protein expression, we noticed that Palbociclib seems to reduce the total protein level of PARP-1. The depletion of components of DNA repair is commonly known to sensitize tumors to treatment with PARP inhibitors, which might explain partially the synergistic interaction between these two targeted inhibitors. This phenomenon was recently shown for lysine-specific methyltransferase 2C (KMTC) in BLCA, whereupon lower expression led to the downregulation of several genes involved in the DNA damage repair and eventually to synthetic lethality with the PARP inhibitor Olaparib [37]. The downregulation of PARP-1 protein levels upon Palbociclib treatment has also been reported recently by others [38]. This confirms our initial finding that overexpression of proteins in the DNA damage pathway confers resistance to Palbociclib.

Hence, one could assume that the addition of Palbociclib should intensify the inhibitory potential of the PARP inhibitor Talazoparib, leading to increased DNA damage and apoptosis. Surprisingly, in combination with Palbociclib, the induction of apoptosis by Talazoparib was significantly reversed. This antiapoptotic potential of Palbociclib has also been shown by others when combined with chemotherapy [39]. However, in contrast to our studies, Yi et al. evaluated a combination therapy approach of CDK4/6 and PARP inhibitors in ovarian cancer and observed an increase in apoptosis under these conditions [36]. This contrary finding might arise from Palbociclib dose-dependent effects. While we applied CDK4/6 inhibitors at a maximum dosage of 1 µM, which displays approximately the tolerated dose in clinical translation, Yi et al. describe the Palbociclib-related induction of apoptosis under concentrations of up to 4 µM. The molecular pattern of triggering apoptosis under such elevated doses of Palbociclib has already been shown. As one consequence, CDK2 is activated, leading to phospho-RAD9 translocation to mitochondria and final reorganization of the Bak/Bcl-xL complex [40].

Cell cycle regulation via cyclin-dependent kinases and the induction of apoptosis are interlinked pathways. The activation of Checkpoint kinase 1 (Chk1) and the stabilization of replication forks seem to require Cyclin E [41]. Similar to factors for DNA repair, cyclin D supports damage response pathways when stably associated with chromatin [42,43]. In our study, Talazoparib treatment of BLCA cell lines resulted in Chk1 activation, while the treatment with Palbociclib led to a depletion of Cyclin E2 and an induction of Cyclin D1. Further, the phospho-RB-dependent exit of senescence activates DNA repair via Nonhomologous End Joining, and its disruption leads to apoptosis [44]. Crucial cell cycle regulators such as RB, p21, or p27 are proteolytic targets of activated Caspase 3 [45,46,47]. In our analysis, PARP inhibition by Talazoparib induced the activation of Caspase 3, induction of p21, and depletion of p27. However, these effects were not further enhanced in the combination and need more detailed analysis.

Palbociclib strikingly decreased the protein level of RB. As monotherapy, Talazoparib enhanced the protein level of pRB and RB in T24 cells, but in the combination, pRB/RB levels were abolished nonetheless. Thus, the role of the RB protein aroused our interest in the therapy combinatory approach.

We identified the expression of functional RB as an essential molecular prerequisite and a presumable predictive marker for the anticancer effect of the Talazoparib/Palbociclib dual therapy. We conclude a crucial role for RB, as its knockdown or expression as a dysfunctional mutant in 5637 cells reversed both, the synergism and the retention of apoptosis induction in the combinatory treatment. CDK4/6 inhibitors depending on RB for therapy response have been reported by several groups, with some exceptions in which cells substitute RB signaling processes by overexpression of FoxM1 [48]. Remarkably, Li et al. describe a synergistic response of concurrent CDK4/6 and PARP inhibition in both RB-proficient and RB-deficient breast cancer cells [35].

For the treatment of urothelial carcinoma with PARP inhibitors, there are several clinical trials ongoing that investigate Olaparib as monotherapy or in combination with, for example, immune checkpoint inhibitors [49]. Individual patients with BLCA showing partial therapy response have been reported within clinical trials of Talazoparib on advanced solid tumors [50,51]. Certainly, one important aspect of clinical trial design is the definition of inclusion criteria based on molecular profiling. Our data and those from other preclinical studies indicate that the expression of functional RB is one stratification marker for inclusion. Additionally, it has been demonstrated that responsiveness to PARP inhibitors is increased in tumors bearing mutations in homologous repair pathways [13].

## 4. Materials and Methods

**Cell lines and cell culture.** Cell lines were maintained as early passages of subconfluent cultures in RPMI or DMEM (Biochrom AG, Berlin, Germany) at 5% or 10% CO_2_, respectively. Media was supplemented with 10% FBS (PAN Biotech GmbH, Aidenbach, Germany), 1% Pen/Strep (Sigma-Aldrich Chemie GmbH, Munich, Germany), and RPMI with an additional 1% NEAA (Biochrom GmbH, Berlin, Germany). UM-UC-3, T24, and HEK 293T were obtained from the American Type Culture Collection (Manassas, VA, USA), RT-112 from the Leibniz Institute German Collection of Microorganisms and Cell Cultures (Braunschweig, Germany), and 5637 were kindly provided by Professor Dr. G. Unteregger (University of Saarland, Homburg/Saar, Germany). Cell lines were kindly authenticated in cooperation with PD Dr. Michèlle Hoffmann (Universitätsklinikum Düsseldorf, Department of Urologie) by short tandem repeat profiling and negatively tested for mycoplasma using qPCR-based Mycoplasmacheck service (Eurofins Scientific SE, Luxemburg).

**Small-molecule inhibitors.** Stock solutions of Olaparib (AZD2281; Selleck Chemicals LLC, Houston, TX, USA) and Talazoparib (BMN683; Selleck Chemicals) were prepared in DMSO. Working concentrations were freshly prepared in medium with controls corresponding to the highest concentrations of DMSO. Palbociclib (PD-0332991) isethionate (Sigma-Aldrich) was dissolved in sterile water.

**Cell viability assay.** A total of 3000–12,000 cells per well were seeded in 48-well plates and after 24 h of incubation treated with respective small molecules. Five days after treatment, cells were fixed using ice-cold 10% trichloroacetic acid (Sigma-Aldrich) and then incubated for at least 1 h at 4 °C. Staining was performed by applying 0.5% (*w*/*v*) Sulforhodamine B (Sigma-Aldrich) in 1% acetic acid (SERVA Electrophoresis GmbH, Heidelberg, Germany) for at least 30 min at room temperature. Culture dishes were then washed with 1% acetic acid and dried completely. The remaining dye was solubilized using 10 mM TRIS (Sigma-Aldrich), and absorbance was measured using a VictorX3 Multilabel Plate Reader (PerkinElmer Inc., Waltham, MA, USA) at 265 nm. Data were analyzed using four-parameter logistic fit and GraphPad Prism V9 software. Data are representative of multiple independent experiments and are presented as the mean ± SD of biological triplicates.

**Chicken chorioallantoic membrane (CAM) assay.** CAM assays were performed as described previously [52]. In brief, 2 million RT-112 Luc cells were seeded on Embryo Day (ED) 9 and topically treated with a single dose of Talazoparib (or DMSO as the vehicle) on ED 11 or once daily from EDs 11 to 14 with Palbociclib. After adding *D*-Luciferin potassium salt (Sigma-Aldrich), luminescence intensity was measured on ED 15 using a Pearl Trilogy Small Animal Imaging System (LI-COR, Inc. Lincoln, NE, USA).

**Statistical analysis.** Treatment conditions in all experiments were compared using *t*-tests in Prism 9 software (GraphPad Software, Inc., San Diego, CA, USA), whereupon the following symbols indicating respective *p*-values: ns: *p* > 0.05, *: *p* ≤ 0.05, **: *p* ≤ 0.01, ***: *p* ≤ 0.001, and ****: *p* ≤ 0.0001.

**Quantification of synergy.** Combination indices (CI) were calculated using CompuSyn (Combo Syn, Inc., Paramus, NJ, USA) according to the Chou–Talalay method [24].

**Immunoblotting.** A total of 6000–8000 cells were seeded in 10 cm dishes and after 24 h of incubation treated with respective small molecules. Four days after treatment, protein lysates were extracted, and protein concentrations were measured using a Pierce BCA Protein Assay Kit (Thermo Fisher Scientific Inc., Waltham, MA, USA) according to the manufacturer’s protocol. Samples were separated using SDS-PAGE and blotted to PVDF membranes using a BioRAD Mini Trans-Blot Cell and Criterion Blotter system. Membranes were then probed using primary antibodies against pChk1, GAPDH, PARP, cl. PARP, Caspase 3, cl. Caspase 3, pRB, Cyclin E2, Cyclin D1, p27, p21 (all Cell Signaling Technology, Inc., Danvers, MA, USA), RB (BD Biosciences Inc., Franklin Lakes, NJ, USA), or γH2A.X (Santa Cruz Biotechnology, Dallas, TX, USA) and subsequent secondary HRP-conjugated antibodies (Dianova GmbH, Hamburg, Germany). Chemiluminescence was measured using a BioRAD ChemiDoc MP imaging system and Amersham ECL Prime Western blotting reagent (GE Healthcare Europe GmbH, Freiburg, Germany).

**Cell cycle analysis.** A total of 50,000–100,000 cells per well were seeded in 6-well plates and after 24 h of incubation treated with respective small molecules. Two to four days after treatment and at 70–80% confluency, cells were fixed in 80% cold ethanol overnight, washed with 1% BSA/PBS, and stained using 4 µg/mL of 7-Aminoactinomycin D (7-AAD, Thermo Fisher Scientific) according to the manufacturer’s instructions. Flow cytometry was performed using a FACSCalibur flow cytometer (BD Biosciences, San Jose, CA, USA), and 10,000 counts of living cells (Gate 1) were measured per sample. Data were analyzed with FlowJo software (FlowJo LLC, Ashland, OR, USA). For determining precise S-phase entry, a Click-it EdU Alexa Fluor 488 flow cytometry assay kit (Life Technologies) was used according to the manufacturer’s protocol. Cells were exposed to EdU for 90 min prior to fixation.

**Apoptosis.** A total of 2000–5000 cells per well were seeded in 96-well plates and after 24 h of incubation treated with respective small molecules. Two to four days after treatment, Caspase-Glo 3/7 (Promega G8091) and Cell Titer-Blue (Promega G8081) assays were conducted in parallel according to the manufacturer’s protocols. Averages of biological triplicates were calculated for each condition and assay separately. Then, each sample was divided by the control of the respective assay (setting vehicle to 100%), and finally, the luciferase samples were divided by the CTB samples (normalization to cell viability).

**Production of lentivirus.** A total of 1.5 million HEK 293T cells were seeded in 10 cm dishes and after 24 h were transfected with 15 µg of psPAX2 (Addgene #12260), 6 µg of pMD2.G (Addgene #12259), and 20 µg of scramble shRNA (Addgene #1864) or pLKO-RB1-shRNA-19 (Addgene #25640) or pLKO-RB1-shRNA-63 (Addgene #25641) using 2.5 M CaCl_2_ and 2× HBS as described previously [53]. The medium was changed 6 h after transfection, and the virus supernatant was collected after 48 h. The supernatant was then sterile filtered with a 0.45 µm syringe filter and stored for a maximum of one week at 4 °C or -80 °C for long-term stocks.

**Transduction of cells.** A total of 0.1, 1.5, and 4 million T24 cells were seeded for viral transduction in 6-well, 10 cm, and 15 cm dishes, respectively. Twenty-four hours after seeding, cells were transduced using 8 µg/mL of polybrene (Sigma-Aldrich). Selection pressure using 1 µg/mL of puromycin was continued for 7 days, starting 24 h after transduction. Passaging of cells and medium change were conducted as necessary. After selection, single clones were expanded.

## 5. Conclusions

Targeted therapies display a superb approach for novel cancer treatment regimens. Here, we emphasize the combinatory treatment of Talazoparib (PARP inhibitor) and Palbociclib (CDK4/6 inhibitor) as a potent therapy approach for BLCA. We further emphasize the expression of the Retinoblastoma protein as a predictive marker for this dual therapy. Thus, stratification of patients in RB-positive should be considered for clinical trial design.

## Figures and Tables

**Figure 1 jpm-11-00340-f001:**
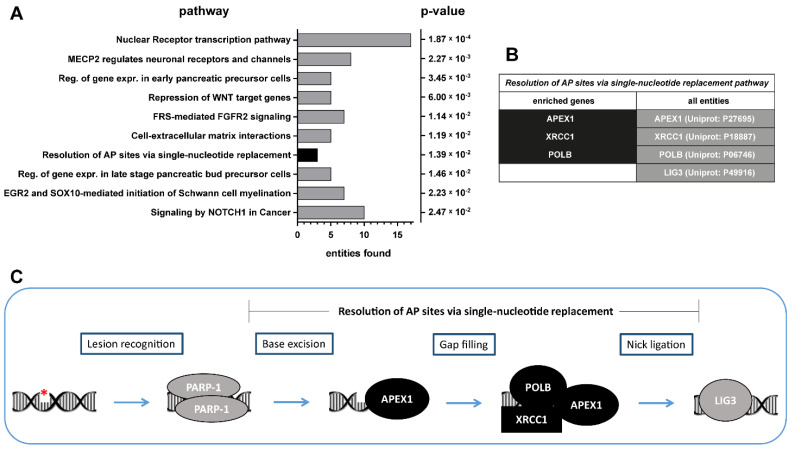
DNA repair pathways confer resistance to Palbociclib treatment. (**A**) The top 10 molecular processes identified by Reactome pathway analysis with sgRNA candidates. (**B**) A comparison of enriched genes conferring resistance to Palbociclib with all entities of the DNA repair process of apurinic/apyrimidinic (AP)-site resolution via single-nucleotide replacement pathway. (**C**) Model of DNA repair via the single-nucleotide replacement pathway. (MECP2: Methyl-CpG-binding protein 2, WNT: Int/Wingless family, FGFR2: Fibroblast growth factor receptor 2, FRS: FGFR substrate, AP: apurinic/apyrimidinic, EGR2: Early growth response protein 2, SOX10: Transcription factor SOX-10, NOTCH1: Neurogenic locus notch homolog protein 1, APEX1: DNA-AP endonuclease, POLB: DNA polymerase beta, XRCC1: X-ray repair cross-complementing protein 1, LIG3: DNA ligase 3).

**Figure 2 jpm-11-00340-f002:**
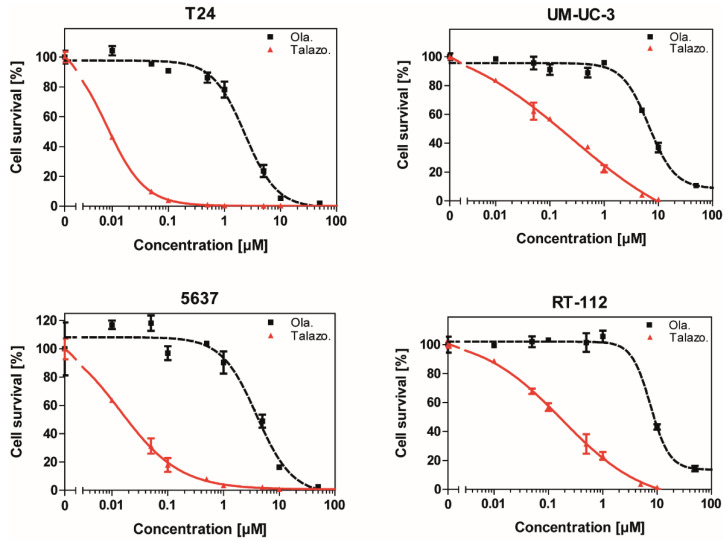
Superior efficiency of Talazoparib compared to Olaparib against bladder cancer cells. Cells were treated for 5 days with Olaparib (Ola.) or Talazoparib (Talazo.), and cell viability was measured using a Sulforhodamin B assay. The datapoint at 0 µM indicates treatment with DMSO as control.

**Figure 3 jpm-11-00340-f003:**
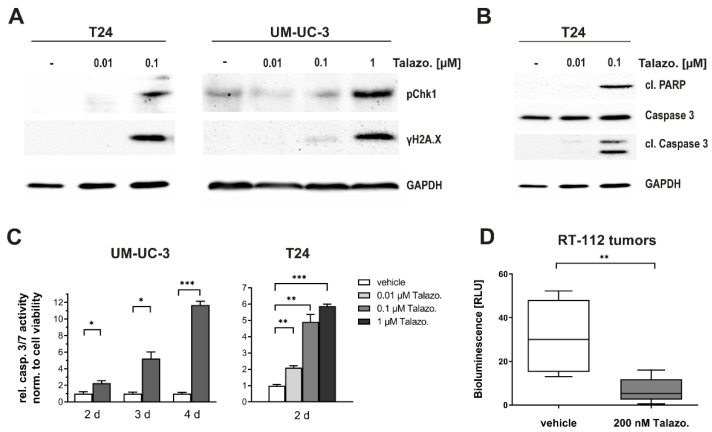
Talazoparib induces both DNA damage and apoptosis and is highly effective in suppressing BLCA tumor growth. (**A**) Protein markers for DNA damage response quantified by Western blot analysis 96 h after treatment. (**B**) Western blot analysis for the quantification of protein markers for apoptosis 96 h after treatment. (**C**) Dependency of Caspase 3/7 activity on the period (UM-UC-3) or the dosage (T24) of Talazoparib treatment. Data are presented as mean ± SD of biological triplicates. (**D**) In vivo effectivity of Talazoparib monotherapy against three-dimensionally grown RT-112 cells by chicken chorioallantoic membrane (CAM) assay. Data are representative of at least three independent experiments. Box and whisker plot depicting the minimum, median, and maximum samples of *n* = 8 for the vehicle and *n* = 14 for Talazoparib-treated tumors. Statistical comparison was performed using *t*-test of GraphPad Prism V9 software, indicating respective *p*-values: *: *p* ≤ 0.05, **: *p* ≤ 0.01, and ***: *p* ≤ 0.001.

**Figure 4 jpm-11-00340-f004:**
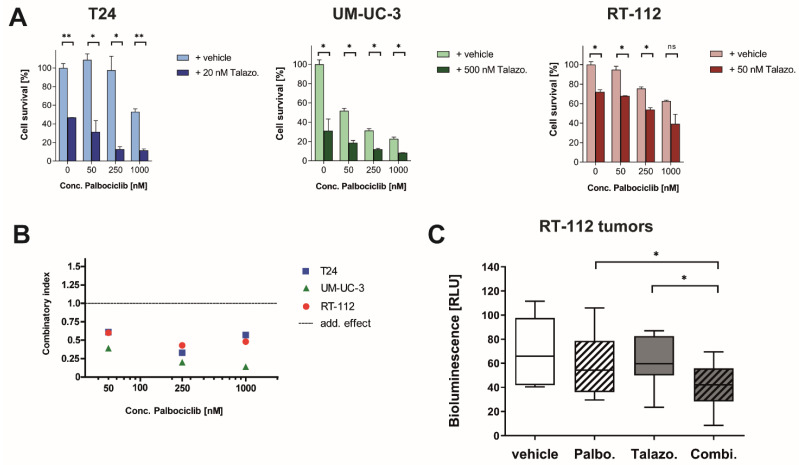
The addition of Talazoparib synergistically enhances the monotherapy of Palbociclib in BLCA models in vitro as well as in vivo. (**A**) Cell viability assay 5 days after treatment; data are representative of six independent experiments and are presented as mean ± SD of biological duplicates. (**B**) Quantification of synergism utilizing cell viability data. Combinatory indices (CI) were calculated using the Chou–Talalay method for drug combination; thereby indicating the following effects: antagonistic (CI > 1), additive (CI = 0), or synergistic (CI < 1). (**C**) CAM assay using a low dose of Talazoparib (50 nM) in combination with Palbociclib (1 µM) to determine in vivo effectivity against three-dimensionally grown RT-112 cells. Data are representative of three independent experiments and are presented as a box and whisker plot depicting minimum, median and maximum samples of *n* = 8 for the vehicle, Talazoparib, and combination, and *n* = 10 for Palbociclib-treated tumors. Statistical comparison was performed using *t*-test of GraphPad Prism V9 software, indicating respective *p*-values: ns: *p* > 0.05, *: *p* ≤ 0.05, and **: *p* ≤ 0.01.

**Figure 5 jpm-11-00340-f005:**
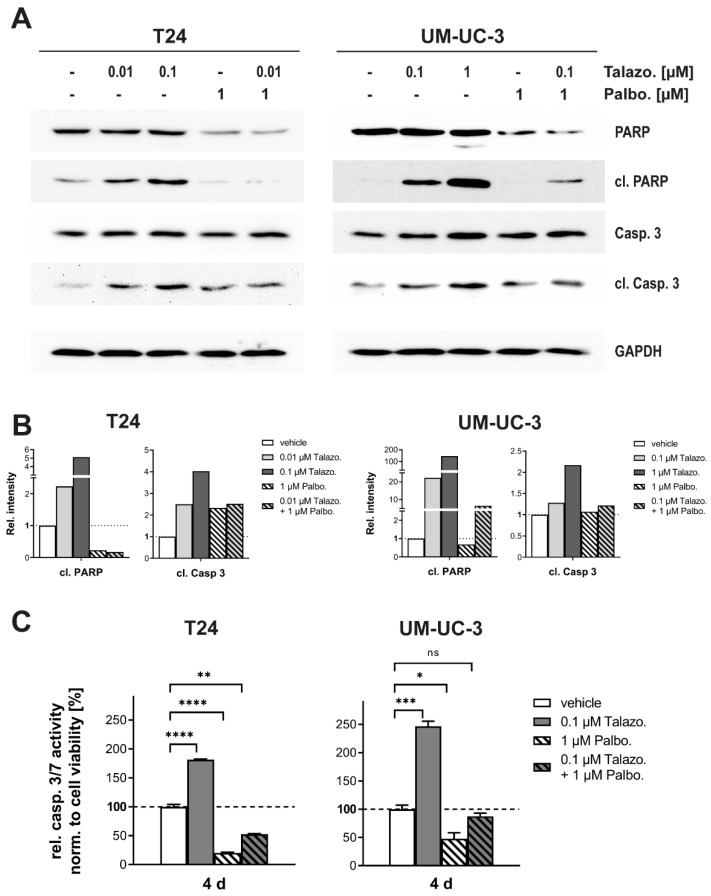
The induction of apoptosis via Talazoparib is reversed in the combinatory treatment with Palbociclib. (**A**) Western blot analysis of protein markers for apoptosis induction 96 h after indicated treatment. (**B**) Quantification of Western blot signals for cleaved products of PARP and Caspase 3. Relative measurement of band intensities was performed using BioRAD ImageLab software. (**C**) Assessment of Caspase 3/7 activity normalized to cell viability 96 h after respective treatments. Data are representative of three independent experiments and are presented as mean ± SD of biological triplicates. Statistical comparison was performed using *t*-test of GraphPad Prism V9 software, indicating respective *p*-values: ns: *p* > 0.05, *: *p* ≤ 0.05, **: *p* ≤ 0.01, ***: *p* ≤ 0.001, and ****: *p* ≤ 0.0001.

**Figure 6 jpm-11-00340-f006:**
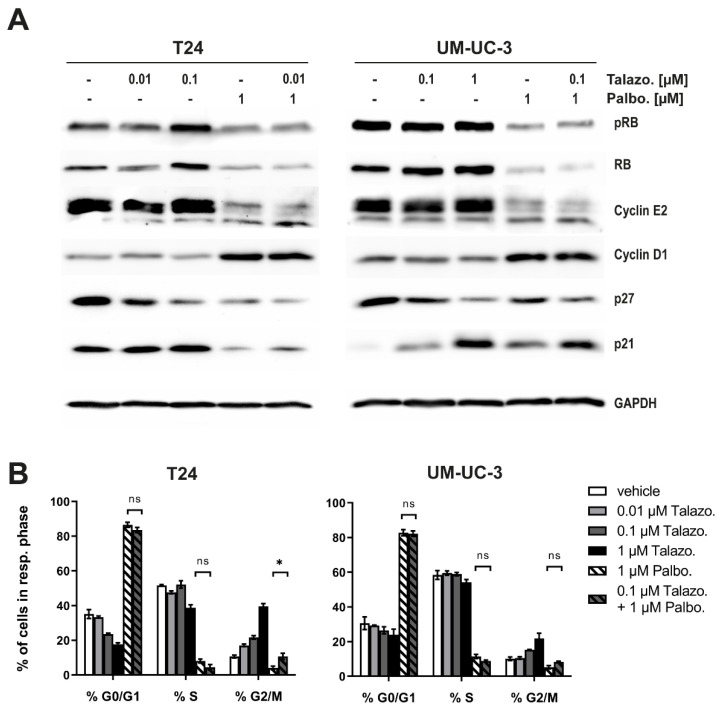
The Talazoparib/Palbociclib combination therapy induces both a G0/G1 and a G2 cell cycle arrest in BLCA cells. (**A**) Key proteins of cell cycle transition were quantified via Western blot analysis 96 h after treatment. (**B**) Analysis of subpopulations for each cell cycle phase 24 h after treatment via flow cytometry after EdU incorporation and staining with 7-AAD. Raw data were analyzed using FlowJo V10 software and are presented as mean ± SD of biological triplicates. Statistical comparison was performed using *t*-test of GraphPad Prism V9 software, indicating respective *p*-values: ns: *p* > 0.05, and *: *p* ≤ 0.05.

**Figure 7 jpm-11-00340-f007:**
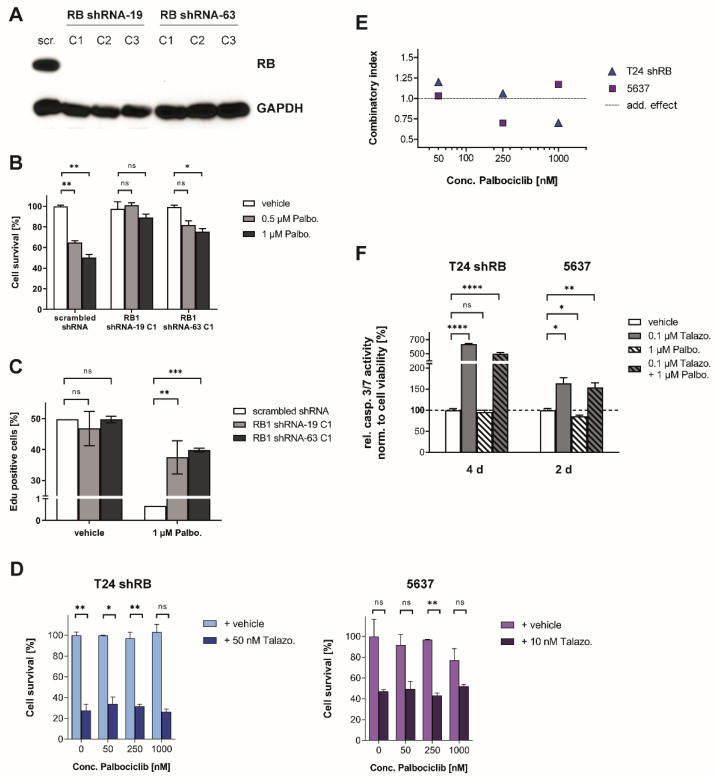
The Knockdown of RB reverses both, the synergism and the retention of apoptosis induction in the Talazoparib/Palbociclib combinatory therapy. (**A**) Western blot analysis of RB expression after lentiviral transduction of the corresponding shRNA. (**B**) Response to treatment with Palbociclib; cell viability measured via Cell Titer-Blue assay 72 h after treatment. (**C**) Flow cytometric analysis after EdU incorporation and 7-AAD staining, 24 h after treatment. (**D**) Cell viability was analyzed via Sulforhodamin B assay 5 days after treatment. (**E**) Quantification of synergism using cell viability data. Combinatory indices (CI) were calculated using the Chou–Talalay method for drug combination. (**F**) Caspase 3/7 activity assay 96 h (T24 shRB) or 48 h (5637) after the respective treatment. Data are representative of three independent experiments and are presented as mean ± SD of biological triplicates. Statistical comparison was performed using *t*-test of GraphPad Prism V9 software, indicating respective *p*-values: ns: *p* > 0.05, *: *p* ≤ 0.05, **: *p* ≤ 0.01, ***: *p* ≤ 0.001, and ****: *p* ≤ 0.0001.

**Table 1 jpm-11-00340-t001:** Overview of IC_50_ concentrations and fold changes when comparing the Poly-(ADP-Ribose) Polymerase inhibitors Olaparib and Talazoparib in bladder cancer cell lines.

Cell Line	IC_50_ Olaparib (nM)	IC50 Talazoparib (nM)	Fold Change (Ola./Talazo.)
T24	2320 ± 320	8.1 ± 0.8	290 ± 70
5637	3970 ± 1010	18 ± 3.4	210 ± 90
RT-112	7610 ± 2260	180 ± 36	43 ± 21
UM-UC-3	6680 ± 630	310 ± 110	22 ± 10

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
