# Peer review of "Combination of Talazoparib and Palbociclib as a Potent Treatment Strategy in Bladder Cancer"

_jpm, 2021, doi:10.3390/jpm11050340_

Round 1

Reviewer 1 Report

Klein and coauthors show that Palbocliclib and Talazoparib displayed synergistic anti-cancer effects on the RB-positive bladder cancer cells. They hypothesize that this synergism is based on complementary G1 and G2 arrest induced by Palbociclib and Talazoparib respectively. The paper addresses an interesting topic and is well written; although some issues should be addressed previous its publication and statistical analysis should be performed for most of the figures.

Regarding Fig 4, where Palbociclin/Talazoparib synergism was evaluated, cell survival values for Talazoparib monotherapy should be included in Fig. 4A panels. I also highly recommend the use of nude mice xenografts for evaluating the in vivo efficiency of Palbociclin/Talazoparib treatment.

In line 178 the authors claim that “Palbociclib as monotherapy does not or only marginally induce cleavage of PARP or Caspase-3”, but in Fig. 5A Palbociclib monotherapy induced similar Caspase 3 cleavage as Talazoparib. Western blot signal should be quantified for proper evaluation of PARP cleavage levels upon Palbociclib treatment. How authors conciliate cl. Caspase 3 inducement upon Palbociclib and Palbociclib/Talazoparib treatments with reduced Caspase3/7 activity when compared with vehicle treated cells?  In addition, are the Caspase3/7 assays performed in fig 5B equivalent to the ones presented in Fig 3C? How % of caspase activity normalized to cell viability is calculated in each figure?

The authors hypothesize that Palbociclib/Talazoparib therapy is probably predicated in a cooperation of respectively induced G1 and G2 arrests by the two inhibitors, but cell cycle data are difficult to interpret as the percentage of G1 arrested cells diminish and increase the percentages of cells in G2 and S phase upon Palbociclib/Talazoparib treatment when compared with Palbociclib treated cells. Data should be complemented with growth curves. Besides, does Figure 6B show one 7-AAD experiment results? The figure should include the media of at least 3 independent experiments.

Regarding Fig. 7, Talazoparib monotherapy values should be added to Fig. 7C and scramble shRNA values to Fig 7F.

Reviewer 2 Report

Reviewer’s report for jpm-1167133

In the article entitled "Combination of Talazoparib and Palbociclib as a potent treatment strategy in bladder cancer", Klein et al have examined the potential use of PARP1 inhibitor Talazoparib and CDK4/6 inhibitor Palbociclib in combination therapy against bladder cancer. They have found that the Palbociclib-induced cell cycle arrest in phase G1, complemented by a G2 block by Talazoparib was highly effective in cells expressing a functional Rb, even though the levels of Talazoparib-induced apoptosis were reduced by Palbociclib.

Reviewer’s comments:

This is a compact, well-designed and interesting piece of work in an important area of carcinogenesis. There are no reservations for the scientific approach or the tools used and procedures followed in this work, whereas the experimental findings justify the conclusions of the article. Therefore, the paper may be published in the Journal of Personalized Medicine after incorporation of some minor changes and correction of a couple of small language mistakes during the editorial process:

  • Figures 3D and 4C

RT112 cells have been chosen for the CAM test. Please, explain why.

  • Figure 4A
    1. In combination treatments, different concentrations of Talazoparib have been used in the three urothelial cell lines used. Please, explain why.
    2. In RT112 cells treated with Palbociclib+Talazoparib, toxicity is not impressive for a 5-day assay. But generally, all cell lines used for Talazoparib and Palbpciclib treatments seem to be in need of a prolonged treatment time in order to obtain significant results. Please, discuss this.

  • Figure 4B

Although T24 and RT112 have significant differences in the individual toxicity tests (Figure 4A), the two cell lines behave similarly in the Chou-Talalay drug combination effect analysis (Figure 4B). Please, explain this.

  • Figure 6A

In cell line UM-UC-3, p21 and p27 responses to Talazoparib treatment follow opposite directions, while it seems that upregulation of p21 here does not conform to the authors’ argument about caspase-3 activation only. Moreover, since both T24 and UM-UC-3 cells are p53 deficient, p21 upregulation must be p53-independent. Please, comment on that.

Reviewer 3 Report

Dear Authors,

In the manuscript titled 'Combination of Talazoparib and Palbociclib as a potent treatment strategy in bladder cancer' the authors have attempted to demonstrate the therapeutic efficacy of a combinatorial treatment of the drugs Talazoparib and Palbociclib in Bladder cancer (BLCA). And also propose, RB expression as a predictive biomarker in the BLCA anti-cancer therapy response. Although the authors have done a very good job in presenting a very convincing set of data to claim that a combinatorial treatment with Talazoparib and Palbociclib drugs do indeed demonstrate a superior efficacy in BLCA, the manuscript suffers from poor English in many sections of the manuscript text and lacks some important components (as described below) that are needed to be added in order to enhance the quality of this manuscript and to be published in JPM:

Major comments: 

  1. The English language grammatical errors in many sections of the manuscript needs to be corrected for a smooth readability of the article. For example, (a) in line 89, 'we are here characterizing....' could instead be, for example, 'here, we have characterized....'. (b) in line number 103, 'against which approved drugs by both, the FDA and EMA, cancer therapy are available' is confusing and grammatically wrong. It could instead be a simple sentence such as 'FDA and EMA approved drugs for cancer therapy are available'. Examples such as these are present in many sections of the manuscript including the results sections as well all of which need to be corrected.
  2. In Figure 3D, authors mentioned two asterisk marks possibly indicating a statistical significance/p value. However, authors failed to add a statistical method description in the methods section and a p value description in the figure legend of Figure 3. Both of which need to be added.  
  3. The choice of using CAM model and the justification for using CAM needs to addressed briefly in the manuscript in the introduction section or in the results section to help the reader understand why CAM has been used specifically instead of mouse models such as NOD-SCID or NSG mice for xenograft studies. 
  4. Figure 3D lacks a y axis label which needs to be added 
  5. What is BLI in figure 4C? In line 164, the authors describe, 'In this model, low dose treatment of the tumors resulted in additive effects that were statistically significant'. This sentence in vague and incomplete. This needs to rewritten to reflect the exact finding that the authors observed in the experiment. Moreover, the figure legends lacks SD and/or p value description of what the asterisk mean exactly? 
  6. All the bar graphs in the figures lack a statistical significance data or a p value which needs to performed and added to the bar graphs. 
  7. The authors need to mention/declare the number of times (n=?) each experiment was performed and the data is representative of what number of experiments, and whether the mean/SD is presented for each experiment in the manuscript??
  8. The western blot in figure 7A is over exposed. Can present a less exposure instead and if there is justification for over exposure, that needs to be explained.
  9. And finally, the manuscript could hugely benefit from adding a model figure that explains the core findings in a nutshell.

Minor comments:

  1. Figure 2 lacks A and B in the figure, but the legend carries A and B which needs to be fixed. The table could be presented separately as a table instead of being a part of figure 2 which might be better.

Round 2

Reviewer 1 Report

The authors satisfactorily replied to all my comments.